# Factors associated with non-specific low back pain in field hockey: A cross-sectional study of Premier and Division One players

**Nick Dobbin** [ID]*, **Craig Getty, Benn Digweed**

Department of Health Professions, Faculty of Health and Education, Manchester Metropolitan University, Manchester, England, United Kingdom

* N.Dobbin@mmu.ac.uk

**Data Availability Statement:** The data that support the findings of this study are available via Dobbin N, Getty C, Digweed B. Factors associated with non-specific low back pain in field hockey: a cross-

## Abstract

### Objective

To determine the extent to which various factors are associated with greater or lesser odds of reporting non-specific low back pain (NS-LBP) in field hockey.

### Methods

To meet the objective of the study, a cross-sectional study design was used with a purposive sampling strategy. A total of 194 responses (~18% of those accessible) from Premier and Division One players within the UK were received using a UK-based online survey. Data collected included information on NS-LBP, participant characteristics, injury history, training related factors, and work and personal factors. The overall and category-specific prevalence of NS-LBP was calculated. Univariable and multivariable logistic regression was used in conjunction with clinical value to identify associations.

### Results

The overall prevalence of NS-LBP was 44.0%, with this varying from 23.5 to 70.0% for categories with responses of "yes" and "no" to experiencing NS-LBP. A total of ten individual factors associated with a greater odds ratio (OR) of reporting NS-LBP (OR = 1.43–7.39) were identified in Premier and Division One players. Five individual factors were associated with reduced odds (OR = 0.11–0.60) of reporting NS-LBP. Seven factors (age, stature, playing position, playing internationally, performing a drag flick, low back stiffness/tightness and occupational factors) were deemed particularly pertinent to those working in field hockey given the magnitude of association and clinical value to clinicians.

### Conclusions

Clinicians working in field hockey can consider the key risk factors identified in this study that are associated with NS-LBP when assessing injury risk, movement screening approaches, and overall athlete management.

sectional study of Premier and Division One players. 2024. Manchester Metropolitan University e-space: https://doi.org/10.23634/MMU.00634825.

**Funding:** The author(s) received no specific funding for this work.

**Competing interests:** The authors have declared that no competing interests exist.

## Introduction

Non-specific low back pain (NS-LBP) is defined as pain, with or without leg pain, between the inferior margin of the twelfth rib and the inferior gluteal folds, and is typically diagnosed by ruling out other causes through patient history and physical examination [1]. NS-LBP is considered the leading cause of disability in the general population, with most people experiencing some form of back pain at least once in their life [2] and many going on to experience chronic low back pain (LBP). In the sporting population, the prevalence of back pain has been synthesised across various reviews [3–5], with resulting highlighting an overall lifetime and point prevalence of 61% and 24%, respectively, across various sporting disciplines. In field-hockey specifically, the overall prevalence is reported to be between 33% and 67% depending on playing status, age and sex [6, 7]. Fett et al. [8] reported slightly higher values amongst 116 field hockey players, with 86% reporting they had experienced LBP over their lifetime (i.e., pain at least once in their life), 82.8% over the last 12 months, 66.4% over the last 3 months, and 44.8% reported LBP during the last 7 days. With the lower back reflecting the fourth most common site of complaint in men's field hockey [9] and being highly prevalent, it's important that those working with these athletes understand the key risk factors associated with NS-LBP, especially considering most field hockey players are amateur, and therefore, must manage work, family, leisure, and sports participation. Failure to identify key risk factors might explain or contribute to the prevalence of NS-LBP, and have a significant financial (e.g., missed work), sporting (e.g., missed games) and psychological (e.g., depression) impact on players.

Research investigating potential risk factors in field hockey is currently limited, but has indicated that higher training loads/volumes, performing a drag flick and drag flick training hours, age, and playing status are associated with an odds ratio (OR) for reporting NS-LBP above 1.0 (OR = 1.024 to 1.564) [10], whilst satisfaction with training and coaches are associated with reduced odds of report NS-LBP (or LBP) (OR = 0.50) [6]. Furthermore, descriptive statistics and between-group comparisons also highlight some positional variation [9] and suggest an influence of eccentric trunk strength and lumbosacral range of motion [11]. However, it's important to note that much of this work is based on descriptive statistics, between-group comparisons or univariate analysis using alpha at 0.05. This means that the *'true'* association between clinically useful (i.e., some might be excluded at 0.05) risk factors, individually, and when accounting for all others, with NS-LBP remains largely unknown.

Beyond field hockey, various other risk factors have been associated with greater odds of NS-LBP (or LBP) including, previous LBP injury/pain, lower-extremity musculoskeletal pain, and higher training volumes and intensities [3–5]. Also, being stressed or tired, occupational exposure (e.g., bending, twisting, and frequently lifting heavy objectives), strenuous workloads, and competing in sport for an extended period have been associated with greater odds of NS-LBP [3, 12–14]. The evidence for age and sex is inconsistent across the literature likely influenced by the groups being assessed, whilst satisfaction with work/sporting, satisfaction with coaches, and being supported during training have been associated with reduced odds [3, 13, 15]. The results highlight that various other factors might be associated with NS-LBP in field hockey but remain largely unexplored. However, such insight might provide important information when considering identification and management of risk as well as screening and training practices.

Given the high prevalence of NS-LBP in field hockey [6, 7], and limited evidence exploring many of the notable risk factors using appropriate analyses (i.e., univariable, multivariable and clinical value of OR) in field hockey, this study aimed to answer the following research question: Are participant characteristics, injury history, training related factors, and work and personal factors associated with greater or lesser odds of experiencing NS-LBP amongst Premier and Division One field hockey players in the UK?

## Material and methods

### Study design and participants

A cross-sectional study design was used to gather information related to NS-LBP from participants purposively recruited as they competed in the men's and women's Premier Division, Division One North and Division One South for field hockey. A total of 62 clubs were identified and sent an invitation. Of those, 18 clubs agreed to share the questionnaire with a potential sample of 1116 participants. A full overview and response rate can be found in Fig 1. Participants were included if they competed in the leagues noted above, and had the ability to, or assistance when, reading and writing in English. Potential participants who were taking medication to manage long-term pain (for any reason) and those with chronic health conditions associated with symptoms of LBP were excluded. Whilst 1116 was estimated to be the maximal number estimated to be available, we determined, using a range of odds ratios from existing

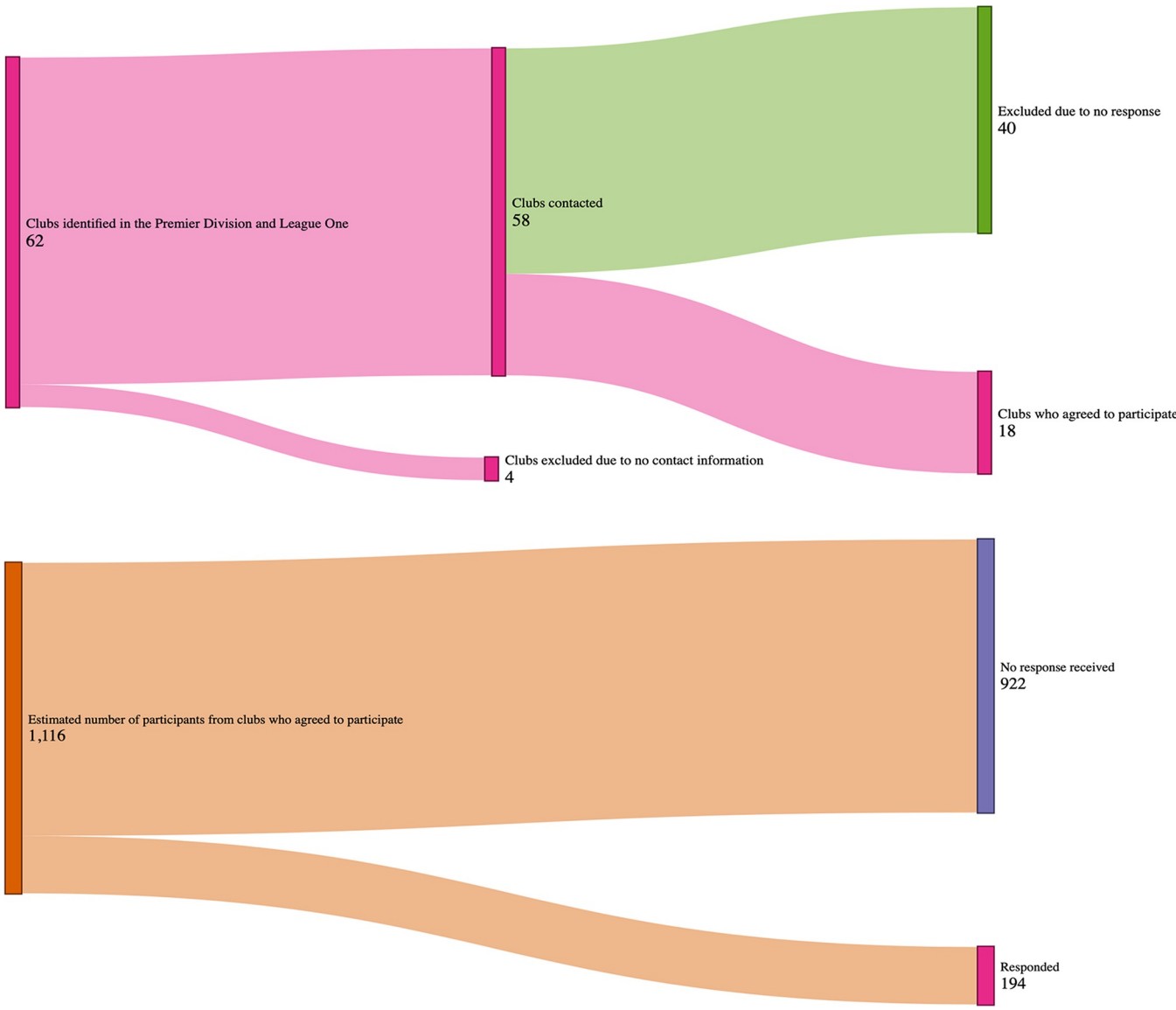

**Fig 1.** Initial identification of clubs (top) and responses from participant (bottom).

research exploring risk factors for LBP (OR range = 0.49 to 5.0) [10, 12, 14, 16] that an 18% response rate would allow for sufficient statistical power using a logistic regression analysis. This sample size estimation was performed on G*Power [17]. Ethics approval was granted for this study by Faculty of Health and Education Research Ethics and Governance Committee at Manchester Metropolitan University (no. 44891). Informed consent was implied by the retuning of the questionnaire as per the institution's consent policy.

## Data collection and questionnaire development

To capture the required data, an online questionnaire was developed by the researchers based on the current evidence on LBP in both the general and sporting population. The questionnaire consisted of the following sections: a participant information sheet, definitions sheet, participant characteristics, injury history, training-related factors, and work and personal factors. All questions resulted in a single option being selection from various categories that were determined *a-priori* using previous literature (e.g., gender, playing positions, stressful occupation), our knowledge of the sport (stick length, drag flick etc.), and clinical reasoning (e.g., surgery, previous episode). The specific categories can be found in Table 1. The questionnaire, once developed, was piloted with several amateur hockey players who focsed on ease of access, logic, question order, suitability of definitions, and terminology. Based on this, several minor logic and terminology changes were made. The responses from the pilot were not included in the dataset for this study. The questionnaire was then distributed (May 2023) to participants via a gatekeeper at those clubs in the United Kingdom who responded to our initial request. They were allowed to distribute the questionnaire by any means they deemed suitable (e.g., club website, WhatsApp groups, monitoring software etc.). Participants were required to access a link via the Joint Information Systems Committee (JISC) survey tool (Jisc, Bristol, United Kingdom). All responses were captured over a 5-week period before it was then closed (June 2023), and the data was exported from JISC to Microsoft Excel (Microsoft® Version 16.59) where it was coded and labelled before being transferred to Statistic Package Social Sciences Version 28 (SPSS, Armonk, USA).

## Statistical analysis

The total number of cases of LBP per risk factor and sub-group were determined along with the prevalence (*Yes LBP/Total per category*) *x* 100 presented an absolute number and percentage, respectively. Each potential risk factor was initially modelled in a univariable logistic regression with LBP (yes vs. no) as the dependent variable to derive the odds ratio (OR), 95% compatibility limits, and statistical probability (presented as absolute values). Then, to avoid ruling out risk factors prematurely, all factors were then entered into a multivariable logistic regression model to determine their contribution when adjusting for all other factors [18]. Once complete, a backward elimination process took place with factors that resulted in probability above 0.05 or an OR between 0.77 and 1.30 removed to provide a final multivariable model [18]. This identified the main independent factors associated with NS-LBP in the sample. The OR were interpreted as: trivial (1.0), small (1.5), moderate (3.5), large (9.0), and very large (>32.0) [19]. The final summary of key risk factors is based on the magnitude of the OR and 95% compatibility intervals and the clinical value to those working in hockey. Statistical analysis was performed using Statistic Package Social Sciences Version 28 (SPSS, Armonk, USA).

## Results

A summary of the recruitment approach and number of responses is provided in Fig 1. A total of 194 responses were received from those competing in the men's and women's Premier

**Table 1. The prevalence of NS-LBP and associated risk factors.**

| Variable | NS-LBP | | Prevalence | Univariable OR (95% CI) | Multivariable OR (95% CI) | Multivariable OR (95% CI) |
|---|---|---|---|---|---|---|
| | Yes (*n*) | No (*n*) | | unadjusted | | Final Model |
| **Participant Characteristics** | | | | | | |
| Age (years) | | | | | | |
| 16–18 | 9 | 29 | 23.7% | $0.39\ (0.15–1.01)^{0.051}$ | $1.52\ (0.28–8.15)^{0.624}$ | $1.43\ (0.30–6.87)^{0.657}$ |
| 19–24 | 26 | 42 | 38.2% | $0.38\ (0.17–0.87)^{0.022}$ | $0.97\ (0.23–4.04)^{0.965}$ | $0.92\ (0.23–3.64)^{0.902}$ |
| 25–30 | 29 | 22 | 56.9% | $1.56\ (0.66–3.70)^{0.315}$ | $3.16\ (0.77–12.89)^{0.110}$ | $3.04\ (0.80–11.53)^{0.102}$ |
| >30 | 22 | 15 | 59.5% | *Ref* | *Ref* | *Ref* |
| Sex | | | | | | |
| Men | 46 | 53 | 46.5% | $1.19\ (0.68–2.11)^{0.541}$ | $0.53\ (0.16–1.80)^{0.309}$ | $0.51\ (0.15–1.67)^{0.264}$ |
| Women | 40 | 55 | 42.1% | *Ref* | *Ref* | *Ref* |
| Stature (cm) | | | | | | |
| <160 | 4 | 8 | 33.3% | $0.45\ (0.12–1.62)^{0.219}$ | $0.11\ (0.01–1.04)^{0.054}$ | $0.11\ (0.01–1.02)^{0.052}$ |
| 161–170 | 12 | 36 | 25.0% | $0.30\ (0.13–0.67)^{0.003}$ | $0.14\ (0.03–0.74)^{0.021}$ | $0.12\ (0.02–0.63)^{0.012}$ |
| 171–180 | 33 | 31 | 51.6% | $0.95\ (0.48–1.87)^{0.881}$ | $0.64\ (0.19–2.19)^{0.476}$ | $0.58\ (0.18–1.86)^{0.354}$ |
| >180 | 37 | 33 | 52.9% | *Ref* | *Ref* | *Ref* |
| Playing experience (years) | | | | | | |
| 0–3 | 0 | 2 | 0.0% | - | - | - |
| 4–7 | 12 | 14 | 46.2% | $1.02\ (0.44–2.40)^{0.960}$ | $0.50\ (0.10–2.51)^{0.402}$ | $0.55\ (0.12–2.59)^{0.452}$ |
| 8–11 | 22 | 30 | 42.3% | $0.87\ (0.45–1.70)^{0.691}$ | $0.41\ (0.12–1.37)^{0.148}$ | $0.43\ (0.14–1.35)^{0.149}$ |
| >11 | 52 | 62 | 45.6% | *Ref* | *Ref* | *Ref* |
| Currently an international player | | | | | | |
| Yes | 23 | 15 | 60.5% | $2.26\ (1.10–4.67)^{0.027}$ | $1.98\ (0.50–7.89)^{0.333}$ | $1.78\ (0.47–6.66)^{0.395}$ |
| No | 63 | 93 | 40.4% | *Ref* | *Ref* | *Ref* |
| Playing position | | | | | | |
| Goalkeeper | 7 | 4 | 63.6% | $1.82\ (0.47–6.99)^{0.383}$ | $1.80\ (0.25–12.89)^{0.556}$ | $1.75\ (0.24–12.76)^{0.579}$ |
| Defender | 29 | 39 | 42.6% | $0.77\ (0.37–1.60)^{0.490}$ | $1.21\ (0.40–3.60)^{0.739}$ | $1.15\ (0.40–3.35)^{0.795}$ |
| Midfielder | 25 | 39 | 39.1% | $0.67\ (0.32–1.40)^{0.285}$ | $1.48\ (0.48–4.56)^{0.499}$ | $1.45\ (0.48–4.34)^{0.507}$ |
| Forward | 25 | 26 | 49.0% | *Ref* | *Ref* | *Ref* |
| Stick Length | | | | | | |
| 35"– 35.5" (86–90 cm) | 10 | 7 | 58.8% | $1.54\ (0.51–4.68)^{0.443}$ | $5.68\ (0.89–36.32)^{0.067}$ | $4.88\ (0.82–28.92)^{0.025}$ |
| 36"– 36.5" (91–93 cm) | 50 | 74 | 40.3% | $0.74\ (0.39–1.42)^{0.374}$ | $1.44\ (0.48–4.23)^{0.517}$ | $1.37\ (0.49–3.82)^{0.550}$ |
| 37"– 37.5" (94–95 cm) | 25 | 27 | 48.1% | *Ref* | *Ref* | *Ref* |
| Drag flick during short corner routines? | | | | | | |
| Yes | 21 | 9 | 70.0% | $3.55\ (1.53–8.24)^{0.003}$ | $3.81\ (1.08–13.42)^{0.037}$ | $4.05\ (1.20–13.74)^{0.025}$ |

(*Continued*)

**Table 1.** (Continued)

| Variable | NS-LBP | | Prevalence | Univariable OR (95% CI) | Multivariable OR (95% CI) | Multivariable OR (95% CI) |
|---|---|---|---|---|---|---|
| | Yes (*n*) | No (*n*) | | unadjusted | | Final Model |
| No | 65 | 99 | 39.6% | *Ref* | *Ref* | *Ref* |
| **Medical History** | | | | | | |
| Number of previous LBP episodes | | | | | | |
| Never | 0 | 108 | 0.0% | - | - | - |
| 1–2 | 51 | 0 | 100.0% | - | - | - |
| 3–4 | 19 | 0 | 100.0% | - | - | - |
| >4 | 16 | 0 | 100.0% | *Ref* | *Ref* | *Ref* |
| Previous lower-back surgery | | | | | | |
| Yes | 5 | 0 | 100% | - | - | - |
| No | 81 | 108 | 42.9% | *Ref* | *Ref* | *Ref* |
| Experience stiffness or tightness in the lower-back the morning after training/match | | | | | | |
| Yes | 57 | 32 | 64.0% | 4.67 (2.54–8.58)$^{<0.001}$ | 4.24 (1.75–10.28)$^{0.001}$ | 3.92 (1.68–9.14)$^{0.002}$ |
| No | 29 | 76 | 27.6% | *Ref* | *Ref* | *Ref* |
| **Training Load** | | | | | | |
| Training hours during a typical week | | | | | | |
| 0–2 | 10 | 9 | 52.6% | 1.48 (0.26–8.50)$^{0.659}$ | 8.02 (0.42–153.52)$^{0.167}$ | 7.39 (0.43–126.64)$^{0.168}$ |
| 3–5 | 41 | 55 | 42.7% | 0.99 (0.21–4.69)$^{0.994}$ | 3.58 (0.23–55.71)$^{0.362}$ | 3.12 (0.23–42.61)$^{0.394}$ |
| 6–8 | 26 | 30 | 46.4% | 1.16 (0.24–5.65)$^{0.858}$ | 1.46 (0.10–20.70)$^{0.781}$ | 1.30 (0.10–16.34)$^{0.842}$ |
| 9–10 | 6 | 10 | 39.1% | 0.80 (0.13–4.87)$^{0.809}$ | 1.07 (0.06–18.76)$^{0.961}$ | 1.12 (0.07–16.90)$^{0.936}$ |
| >10 | 3 | 4 | 42.9% | *Ref* | *Ref* | *Ref* |
| Hockey matches/week | | | | | | |
| 1 | 48 | 80 | 37.5% | 0.90 (0.15–5.58)$^{0.910}$ | 0.84 (0.02–31.32)$^{0.922}$ | 0.87 (0.02–34.89)$^{0.940}$ |
| 2 | 36 | 25 | 59.0% | 2.16 (0.34–13.88)$^{0.417}$ | 5.34 (0.14–208.17)$^{0.370}$ | 5.48 (0.13–226.35)$^{0.371}$ |
| ≥3 | 2 | 3 | 40.0% | *Ref* | *Ref* | *Ref* |
| Frequency of lower-back or core strengthening exercises during a typical week | | | | | | |
| Never | 31 | 36 | 46.3% | 0.70 (0.29–1.66)$^{0.418}$ | 1.22 (0.32–4.61)$^{0.768}$ | - |
| 1–2 times | 38 | 59 | 39.2% | 0.47 (0.21–1.08)$^{0.076}$ | 1.02 (0.28–3.73)$^{0.973}$ | - |
| ≥3 | 17 | 13 | 56.7% | *Ref* | *Ref* | *Ref* |
| Participation in the indoor hockey season | | | | | | |
| Yes | 27 | 59 | 31.4% | 1.90 (0.98–3.67)$^{0.057}$ | 0.98 (0.34–2.80)$^{0.964}$ | - |
| No | 21 | 87 | 19.4% | *Ref* | *Ref* | *Ref* |
| **Occupational and Personal Factors** | | | | | | |
| Lifting heavy loads at work | | | | | | |
| Yes | 23 | 10 | 69.7% | 3.58 (1.60–8.02)$^{0.002}$ | 2.99 (0.85–10.56)$^{0.087}$ | 2.53 (0.80–8.00)$^{0.113}$ |
| No | 63 | 98 | 39.1% | *Ref* | *Ref* | *Ref* |

*(Continued)*

**Table 1.** (Continued)

| Variable | NS-LBP | | Prevalence | Univariable OR (95% CI) | Multivariable OR (95% CI) | Multivariable OR (95% CI) |
|---|---|---|---|---|---|---|
| | Yes (*n*) | No (*n*) | | unadjusted | | Final Model |
| Standing >30 minutes/hour during work | | | | | | |
| Yes | 52 | 56 | 48.1% | $1.42 (0.80–2.52)^{0.231}$ | $0.97 (0.29–3.24)^{0.960}$ | - |
| No | 34 | 52 | 39.5% | *Ref* | *Ref* | *Ref* |
| Sitting >30 minutes/hour during work | | | | | | |
| Yes | 45 | 62 | 42.1% | $0.81 (0.46–1.44)^{0.480}$ | $1.27 (0.40–3.98)^{0.697}$ | - |
| No | 41 | 46 | 47.1% | *Ref* | *Ref* | *Ref* |
| Perceive work to increase fatigue | | | | | | |
| Yes | 50 | 56 | 47.2% | $1.29 (0.73–2.28)^{0.383}$ | $0.63 (0.23–1.74)^{0.369}$ | $0.60 (0.23–1.59)^{0.306}$ |
| No | 36 | 52 | 40.9% | *Ref* | *Ref* | *Ref* |
| Perceive work to prevent recovery | | | | | | |
| Yes | 45 | 28 | 61.6% | $3.14 (1.72–5.73)^{<0.001}$ | $2.36 (0.83–6.74)^{0.107}$ | $2.03 (0.75–5.52)^{0.164}$ |
| No | 41 | 80 | 33.9% | *Ref* | *Ref* | *Ref* |
| Perceive sleep as good quality | | | | | | |
| Yes | 54 | 91 | 37.2% | $0.32 (0.16–0.62)^{0.001}$ | $1.13 (0.34–3.78)^{0.844}$ | - |
| No | 32 | 17 | 65.3% | *Ref* | *Ref* | *Ref* |
| Perceive sleep as good quantity | | | | | | |
| Yes | 49 | 81 | 37.7% | $0.44 (0.24–0.81)^{0.009}$ | $0.46 (0.16–1.31)^{0.146}$ | $0.50 (0.21–1.19)^{0.117}$ |
| No | 37 | 27 | 57.8% | *Ref* | *Ref* | *Ref* |
| Have you ever had periods of job dissatisfaction? | | | | | | |
| Yes | 40 | 43 | 48.2% | $1.31 (0.74–2.33)^{0.349}$ | $0.76 (0.30–1.89)^{0.550}$ | - |
| No | 46 | 65 | 41.4% | *Ref* | *Ref* | *Ref* |
| Hockey performance dissatisfaction | | | | | | |
| Yes | 45 | 60 | 42.9% | $0.88 (0.50–1.55)^{0.654}$ | $0.67 (0.27–1.65)^{0.378}$ | - |
| No | 41 | 48 | 46.1% | *Ref* | *Ref* | *Ref* |
| Perception of job stress | | | | | | |
| Never stressful | 4 | 13 | 23.5% | $0.17 (0.05–0.65)^{0.009}$ | $0.10 (0.01–0.88)^{0.038}$ | $0.13 (0.02–0.99)^{0.048}$ |
| Rarely stressful | 13 | 28 | 31.7% | $0.26 (0.10–0.68)^{0.006}$ | $0.33 (0.08–1.13)^{0.116}$ | $0.32 (0.08–1.22)^{0.095}$ |
| Sometimes stressful | 46 | 54 | 46.0% | $0.48 (0.22–1.06)^{0.068}$ | $0.33 (0.100–1.14)^{0.080}$ | $0.35 (0.11–1.14)^{0.083}$ |
| Frequently stressful | 23 | 13 | 63.9% | *Ref* | *Ref* | *Ref* |
| Stressful life event | | | | | | |
| Yes | 49 | 41 | 54.4% | $2.16 (1.26–3.86)^{0.009}$ | $1.78 (0.76–4.21)^{0.185}$ | $1.62 (0.71–3.72)^{0.255}$ |
| No | 37 | 67 | 35.6% | *Ref* | *Ref* | *Ref* |

Note: LBP = low back pain. LB = low back. OR = Odds ratio. Ref = reference/referent category. Superscript number represent the probability statistic. Multivariable model included all factors to determine the effect each factor had when fully adjusted. The final model included all factors that deemed to have an association (using OR and P value) or clinical value.

Division, Division 1 North and Division 1 South. This reflects an estimated response rate of 18% from the clubs contacted.

The responses reflected a near-equal representation of male ($n$ = 99) and female ($n$ = 95) participants split across four broader age groups (16–18 years = 38, 19–24 years = 68, 25–30 years = 51, >30 years = 37). Almost one third of respondents fell into one of the three categories of stature with those <160 cm being somewhat underrepresented in the sample; a similar pattern was reflected for stick length. Most athletes had considerable experience playing field hockey, with almost 60% having over 11 years of playing experience. Goalkeepers were the smallest represented group as expected given the number of positions available; all others were equally represented in the responses. Finally, 20% of our sample were, at the time of completing the questionnaire, competing internationally.

Forty-four percent of all participants who completed the questionnaire had reported previously experiencing low-back pain with 1–2 episodes being most common as well as stiffness or tightness in the lower back the morning after playing hockey. Results for training load suggested that most participants completed between 3 and 8 hours of training during a typical week as well as 1–2 matches. Whilst some respondents suggested they completed specific lower-back or core strengthening, 35% did not. Occupational and personal factors generally indicated equal responses for most, except for work preventing recovery, sleep quality, and sleep quantity (see Table 1).

Characteristics that were associated with a higher prevalence of NS-LBP concerned older players (>25 years), taller players (>171 cm), and those competing at an international level. Also, individuals using a shorter hockey stick (35"– 35.5") or who complete a drag flick during short corner routines reported a considerably higher prevalence of NS-LBP. Experiencing tightness or stiffness after training/match-play, perceiving work to prevent recovery and be stressful, having poor sleep quality/quantity, and lifting heavy load in work were associated with a higher prevalence of NS-LBP.

Results from the univariable analysis indicated that 13 individual factors were associated (not all "significant") with greater odds of NS-LBP. These included being between the ages of 25–30 years, being a goalkeeper, competing internationally, playing with the shortest stick, performing a drag flick, reporting lower-back stiffness or tightness following hockey, having the lowest weekly training load, playing two matches per week, competing in indoor hockey, lifting heavy loads, standing for at least 50% working time, perceiving work to impact recovery, and experiencing a stressful life event. In contrast, 5 factors were associated with lower odds of NS-LBP. Specifically, being in the younger age-groups (<25 years), being of less stature (<170 cm), being a mid-fielder, reporting sleep quality and quantity as good, and having a less than frequently stressful job were associated with lower odds compared to the referent/reference group. The odds ratio and inferential statistical interpretation is presented in Table 1.

In the multivariable model, two additional factors were included from the univariable analysis, namely sex and playing experience. Being in the younger age groups was no longer associated with lower odds, demonstrating a degree of instability for this group (e.g., 16–18 years) but not 25–30 years. Similarly, the odds of NS-LBP in midfields and those perceived their sleep the be good quality suggest some instability going from lower to higher odds between the univariable and multivariable analyses. The opposite was found for perceiving work to increase fatigue. Four factors were associated with lower odds of NS-LBP including sex (male), being less experienced that those with >11 years, perceiving sleep to be of good quantity, and experiencing less stress at work. Of the original 13 factors associated with greater odds of NS-LBP, 10 remained (Table 1).

The final model indicated that age, playing at an international level, use of a smaller stick, partaking in the drag flick, experiencing stiffness/tightness after hockey, training 0–2 and 3–5

**Table 2. Summary of key risk factors associated with NS-LBP.**

| Variable | Interpretation |
|---|---|
| Age | Players aged 25 to 30 are at greater odds (OR = 3.04) of developing NSLBP compared to the older group. |
| Stature | Players under <170 cm were at lower odds (OR = 0.11 to 0.12) of developing NSLBP compared to those >170 cm. |
| Position | Goalkeepers and midfielders seem to present with slightly higher odds of NSLBP (OR = 1.45 to 1.75). |
| Playing International Hockey | Those competing at an international level are at higher odds of LBP (OR = 1.78). |
| Drag Flick | Those who drag flick in short corner routines are at 4.05 times greater odds of developing NSLBP. |
| LB stiffness or tightness | Players who perceive their LB to be stiff or tight are at 4.82 times greater risk of developing NSLBP |
| Occupational factors | Lifting heavy load at work, perceived work to prevent recovery, and experiencing a stressful life event were associated with greater odds of LBP (OR = 1.62 to 2.53). Perceiving work to increase fatigue, good sleep quantity, and minimal job stress were associated with lower odds of LBP (OR = 0.13 to 0.60). |

hours per week, competing in two matches per week, lifting heavy objects at work, perceiving work to impact recovery, and experiencing a stressful life event were associated with greater odds ranging from 1.43 to 7.39 (Fig 2). In contrast, being a male, being of smaller stature, perceived work to increase fatigue, perceiving sleep to be good quantity, and experiencing less frequent job stress were associated with lower odds of NS-LBP (OR = 0.10–0.60). A summary of risk factors associated with NS-LBP based on statistics, the size of the OR, and our own clinical interpretation is presented in Table 2.

## Discussion

The aim of this study was to identify the extent to which various participant characteristics, medical factors, training-related factors, and work and personal factors are associated with NS-LBP amongst Premier and Division One field hockey players. The results from this study indicate an overall lifetime prevalence of 44%, and that increased age, stature, playing position, competing internationally, performing a drag flick, experiencing stiffness/tightness, and occupational factors influenced the odds NS-LBP. The latter included three factors associated with greater odds (i.e., lifting heavy loads, work preventing recovery, and experiencing a stressful life event) and three (i.e., work increasing fatigue, good sleep quantity and minimal job stress) were associated with reduced odds of NS-LBP.

The lifetime prevalence of NS-LBP in this study agreed with that previously reported in field hockey (~56%) [6, 7] and across various other sports such as cycling (~55.1%) [15], ice hockey (60.1%) [20], basketball (~ 46–53) [3], and soccer (~42%) [21]. It was, however, lower than the pooled estimate (prevalence = ~ 63 to 77%) across multiple sports [3–5]. This difference may reflect the high heterogeneity caused by the sports included, definitions of LBP, and methodological differences. The lower prevalence in this study might also reflect the unstated period when answering questions about experience NS-LBP in this study, thus potentially reflecting points closer to the period the questionnaire was completed rather than their lifetime, especially since point prevalence is generally lower 12 months or lifetime prevalence [8]. Nonetheless, a prevalence of 44% suggests NS-LBP is a concern in field hockey and that players, coaches, and medical personnel should be aware of potential risk factors (e.g., sport-specific tasks, movement demands, work-related factors, previous injury etc.) and implement screening, training and management practices to support athletes.

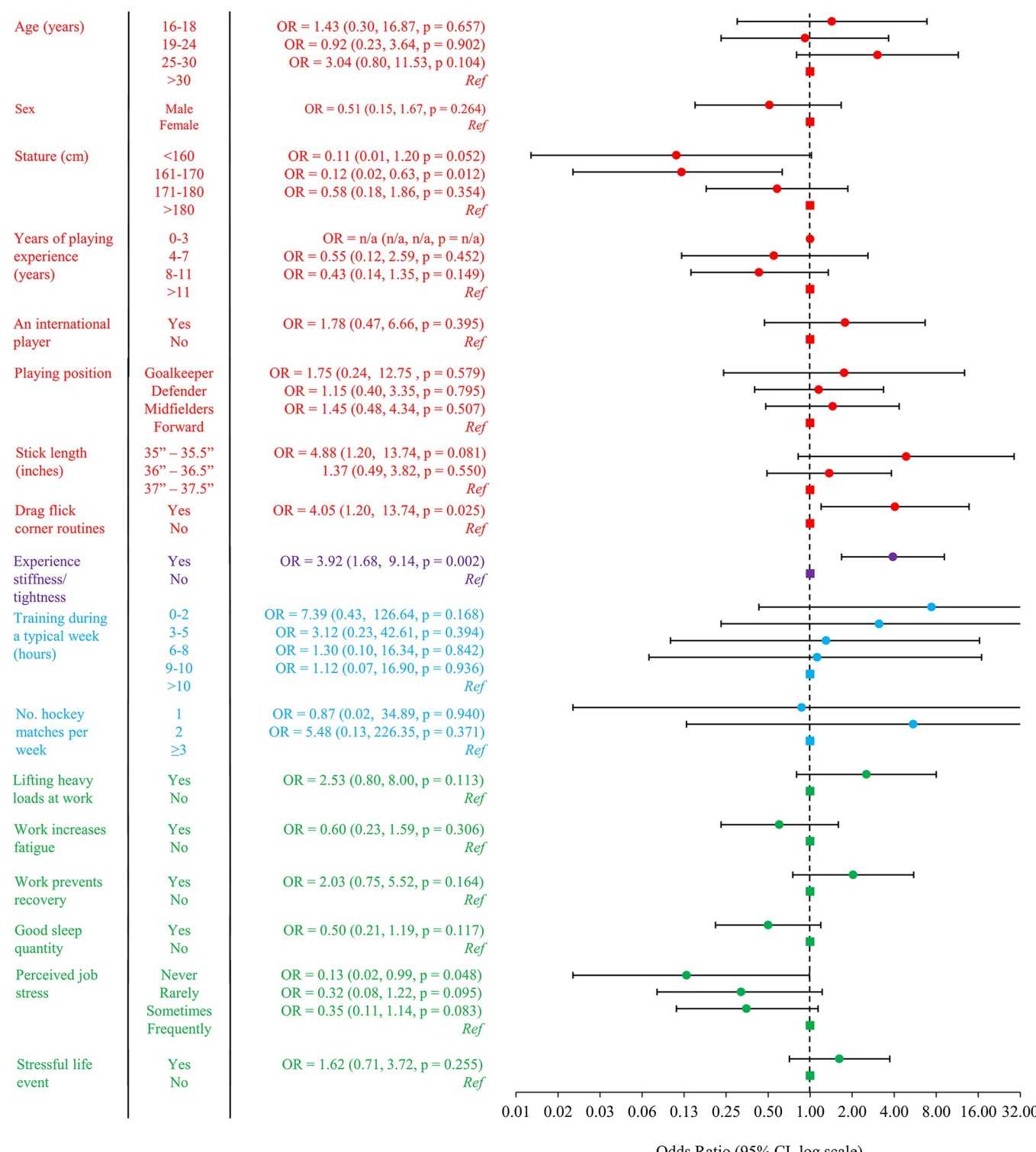

**Fig 2. Association between independent factors with NS-LBP in the final multivariable model.** Note: OR = Odds ratio. Ref = reference/referent category (squares). CI = compatibility limits.

This study found several individual characteristics that affect the odds of NS-LBP. Our findings provided a mixed picture on the influence of sex on NS-LBP whereby males demonstrated a slightly higher prevalence than females yet a lower OR in the final model once accounting for various other factors. The results for prevalence largely agree with Trompter et al. [3] who reported six studies support male athletes (court-based or weight-category sports) reporting a higher prevalence of NS-LBP compared to female athletes. This does, however, contrast the results on Fett et al. [8] who surveyed 1114 athletes across various sports. Fett et al. [8] reported a higher prevalence of NS-LBP in female athletes over a 7-day and 3-month period, but not 12 months or the athlete's lifetime. Similarly, van Hilst et al. [6] reported that young female field hockey players reported a higher prevalence than their male counterparts due to potentially being better able to tolerate the high training and mechanical loads [3]. The point estimate in the final model within this study suggests lower odds of reporting NS-LBP in male field-hockey players, though we do acknowledge that positive, negative and null associations are encompassed within the compatibility intervals. As such, our study provides no further clarity on the existence of a sex-difference in NS-LBP within field hockey, with much of the variance explained by other factors included in our final model (e.g., stature, drag flick routine, stiffness/tightness) and that high within-sex variability exists.

In the general population, there is agreement that the prevalence and odds of reporting LBP increase with age; however, this relationship has only tentatively been noted in athletic populations due to the narrow ages range included [3, 4, 5, 8]. Indeed, Fett et al. [8] reported a positive correlation between age and back pain with lifetime prevalence increasing by 12% across an age difference of ~17 years whilst Wall et al. [4] reported on a ~10% increase in prevalence from 11–13 years to 14–17 years. That said, when considering individual studies, including a study in field hockey [6], most find no association. In this study, there was a suggestion that those aged 25 to 30 years had higher odds of reporting NS-LBP at both a uni- and multivariable level when compared to the > 30 years age group, though we urge some caution given the width of the compatibility intervals. It is possible that this age group reflects a period of where we see a transition into a young adult, movement into formal employment, and a higher proportion experiencing high training loads, take on positions of responsibility, competing internationally, or playing in the goalkeeper and midfield position.

Our data support the notion that those training and competing in field-hockey are a higher risk of NS-LBP due to sport-specific features including use of a hockey stick and the drag flick corner routine as well as overall training and competition load. We observed an almost five times greater odds of reporting NS-LBP in those using the shortest hockey stick (35.0" to 35.5") and four times greater odds for those completing a drag flick corner routine. Both features of the sport are likely to increase the odds of NS-LBP due to the hyperflexed position of the trunk during bending, twisting and turning under load, and awkward body positioning during other activities (e.g., dribbling, contact with an opponent) [6, 8, 12]. In contrast to this, Haydt et al. [7] suggested there was no influence of posture on LBP given there was minimal difference in prevalence between a group of 90 female NCAA Division III players and an age-match control group. However, it must be noted that this study contained numerous methodological and statistical issues meaning caution is needed, and that is it likely the shorter stick and drag flick routine can increase the odds of NS-LBP in field-hockey. We also observed that those undertaking the least training hours per week and two matcher per week had greater odds or reporting NS-LBP which generally agrees with the lower end of the U-shaped relationship between risk and training. Collectively, our results suggest that there is a need to ensure field-hockey players are exposed to sufficient training to elicit adaptation, such as strength or lumbar range of motion [6, 11], but moderated when it comes to the drag flick corner routine.

Further, correct fitting of a hockey stick and further research into the biomechanics of key sporting actions involving the hockey stick is warranted.

A previous episode of LBP is commonly associated with greater odds of reporting NS-LBP [3, 5, 22], however we were unable to document this in our study as there was insufficient representation for comparator groups. A similar observation was apparent for previous surgery. We did, however, observe an association between stiffness/tightness in the lower-back the morning after hockey activity with reporting of NS-LBP at both a uni and multivariable level. It is possible that this association is explained by compensatory movement during everyday or sporting activity, poor posture, joint dysfunction, decreased flexibility and muscle imbalance–all of which likely contribute to NS-LBP risk. Clinicians working with field hockey players ought to consider suitable physiotherapy modalities for maximising range of motion, strengthening the core, and recovery post training/competition to moderate the risk of NS-LBP [6, 11].

Consideration for occupation and personal factors is yet to be explored across many sports including field hockey. Our results suggest there are two key factors worth considering when working with field hockey players. Firstly, given most field hockey players are amateur, manual tasks that involve lifting heavy items at work was associated with greater odds of NS-LBP—a finding often observed in low wage workers or those in primary clinical settings [12, 14]. Secondly, job stress or a stressful life event increased the odds of reporting NS-LBP in this study in accordance with psychological construct in the multidimensional sport injury risk profile in athletes [23]. Whilst exploring the exact mechanisms is beyond the scope of this study, it is well-reported in non-athletic populations that stress (in various forms) is associated with LBP [24]. It is suggested that the mechanisms for this centre around neuroendocrine and psychophysical responses that cause a heightened perception of pain and processing of pain signals [24]. Athletes are not exempt to these stressors and may have to deal with additional stress associated with sport (e.g., coaches, selection, results, expectations and pressure) [25]. As such, it is important for those working with field hockey players to be cognisant to stress that might be experienced by a field hockey player both within and outside of the sport given stress can trigger and amplify pain [24]. Beyond stress, other occupational and personal factor demonstrated less certainly, though did potentially indicate that sleep quantity and having an occupation that limits fatigue might be an important consideration when working with field hockey players [26]. For example, Wami et al. [14] noted that housekeeping staff who took more than 3 rest breaks per day had 51% lower odds of LBP compared to those having less than 30 minutes rest per day, whilst Steffens et al. [12] reported that being fatigued and tired increase the risk substantially (OR = 3.3 to 4.5).

## Limitations and conclusions

This study it is not without its limitations. Firstly, whilst we feel 194 is an adequate sample overall based on our power calculation and approach (P, values OR and clinical value), we do acknowledge that some groupings are small (e.g., < 160 cm) suggesting a reduced precision around the point estimate and higher type II error rate. We are also unable to comment if all 10 clubs who responded contributed to the responses as we sought to protect the identity of the club, so did not ask this question. We accept that not all potential risk factors were explored such as satisfaction with performance or coaches [6], technical ability, and key movement patterns, amongst others. Finally, we highlight that we are unable to comment on the association with previous episodes of LBP as this failed to reflect a precise time-period compared the question used to derive prevalence. That said, we feel this study offers some important conclusions for clinicians working in the sport of field hockey who are likely to see NS-LBP within their

squad or clubs with a prevalence of ~44%. Clinicians are also likely to have players who present with individual or a combination of risk factors that are associated greater or reduced odds of NS-LBP in hockey. Further work might consider expanding on the risk factors in this study as well as seeking to devise screening or preventative measures to moderate the risk of NS-LBP in field hockey as well as rehabilitation strategies should LBP present like that observed in rowing [27].

## Acknowledgments

We would like to thank the participants who took the time to complete the questionnaire.

## Author Contributions

**Conceptualization:** Nick Dobbin, Craig Getty, Benn Digweed.

**Data curation:** Craig Getty.

**Formal analysis:** Nick Dobbin.

**Investigation:** Craig Getty.

**Methodology:** Nick Dobbin, Craig Getty, Benn Digweed.

**Supervision:** Nick Dobbin, Benn Digweed.

**Visualization:** Nick Dobbin.

**Writing – original draft:** Nick Dobbin, Craig Getty, Benn Digweed.

**Writing – review & editing:** Nick Dobbin, Craig Getty, Benn Digweed.

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
