## [Decision Letter · Decision Letter 0]

30 Apr 2024

PONE-D-24-12011Factors associated with non-specific low back pain in field hockey: a cross-sectional study of Premier and Division One players.PLOS ONE

Dear Dr. Dobbin,

Thank you for submitting your manuscript to PLOS ONE. After careful consideration, we feel that it has merit but does not fully meet PLOS ONE’s publication criteria as it currently stands. Therefore, we invite you to submit a revised version of the manuscript that addresses the points raised during the review process.

We look forward to receiving your revised manuscript.

Kind regards,

Shazlin Shaharudin

Academic Editor

PLOS ONE

Journal Requirements:

**Additional Editor Comments:**

Please revise the manuscript according to the feedback from the reviewers noting that you don't have to cite the specific reference(s) as mentioned by one of the reviewers.

Reviewers' comments:

Reviewer's Responses to Questions

**Comments to the Author**

1. Is the manuscript technically sound, and do the data support the conclusions?

Reviewer #1: Yes

Reviewer #2: Yes

2. Has the statistical analysis been performed appropriately and rigorously? 

Reviewer #1: Yes

Reviewer #2: Yes

3. Have the authors made all data underlying the findings in their manuscript fully available?

Reviewer #1: No

Reviewer #2: No

4. Is the manuscript presented in an intelligible fashion and written in standard English?

Reviewer #1: Yes

Reviewer #2: Yes

5. Review Comments to the Author

Reviewer #1: Given the study focuses solely on athletes, it would be more accurate to use a consistent term like "athlete" or "player" throughout the text instead of "participants."

Abstract

Line ‘Data collected include information on NS-LBP and participant characteristics, injury history, training related factors, and work and personal factors were obtained.’, here, omit the words ‘were obtained’.

Introduction

To provide the most up-to-date picture of low back pain (LBP) prevalence in different sports, it would be beneficial to update the introduction and discussion sections on prevalence with findings from a recent study by Ansari and Sharma (2023) [Ansari, S., & Sharma, S. (2023). Prevalence and risk factors of chronic low back pain in university athletes: a cross-sectional study. The Physician and Sportsmedicine, 51(4), 361–370. https://doi.org/10.1080/00913847.2022.2108351] on chronic LBP of non-specific originin university athletes. Additionally, incorporating risk factors explored in this research would strengthen the introduction's overview of relevant studies on the topic.

There seems to be an error in citation number eight. The author's name is listed as Frett, but it should be Fett. Please update all references and the bibliography entry accordingly

Material and methods

It is recommended to use sub-headings in this section, example, ethical considerations and study registration, study setting and design, participants/sample size/eligibility criterion, data collection/questionnaire, statistical analyses, etc.

Were the data from the pilot testing version included in the present study?

What measures were used to depict the data, mean (SD)/ n (%), add details of these in the statistical analyses part.

How were the variables fit into a univariate regression, was a significant correlation test was used to ascertain its inclusion in the univariate regression?

Add details in the statistical analyses section on how the categorization of the variables was done for analyses.

Results

It is recommended to provide some of the basic demographic data in terms of mean (SD).

Discussion

Lines 260-271: Add findings from the studies:

Moradi V, Memari A-H, and ShayestehFar M, et al. Low back pain in athletes is associated with general and sport specific risk factors: a comprehensive review of longitudinal studies. Rehabil Res Pract. 2015;2015 2015:850184 . DOI:10.1155/2015/850184.

Wilson F, Ardern CL, Hartvigsen J, et al. Prevalence and risk factors for back pain in sports: a systematic review with meta-Analysis. Br J Sports Med. 2021;55(11):601–607.

Lines 318-323, To understand how sleep might influence LBP risk across different sports, it's worth including sport-specific results from the below mentioned study on sleep and LBP:

Ansari, S., & Sharma, S. (2024). Sleep Status and Chronotype in University Athletes with and without Chronic Low Back Pain: A Cross-Sectional Study. Sleep Science.

Reviewer #2: Abstract: in method, please state study design, sampling method. Include how many club involved for 194 participants

keyword: add low back pain

Introduction: line 100 after field hockey (ref)

materials and method:

Please add exclusion criteria

Please specify the statistical test

Result: please include P value in Table 1

Table 1= referent or reference?

line 168-169 the sentence is not clear

6. PLOS authors have the option to publish the peer review history of their article (what does this mean?). If published, this will include your full peer review and any attached files.

Reviewer #1: No

Reviewer #2: No

---

## [Author Response · Author response to Decision Letter 0]

29 May 2024

Response to Reviewers 

General Comments 

We thank the reviewers for their time and effort reviewing our manuscript. We are pleased that you both felt we have produced a technically sound paper that is well-written and used appropriate statistical analysis. In response to the “No” concerning data availability, we do intend to provide a DOI to the data upon acceptance. This was stated in our initial submission. 

We provide a point-by-point response below to the valuable feedback you have provided. 

Thank you 

Reviewer #1: 

Given the study focuses solely on athletes, it would be more accurate to use a consistent term like "athlete" or "player" throughout the text instead of "participants."

We thank the reviewer for their feedback here. We would prefer to use the term participants to ensure we are in accordance with the phrasing used in the PLOSone guidelines. Also, we do highlight that the term participant is generally preferred by those who are voluntarily participating in a research study. 

Abstract

Line ‘Data collected include information on NS-LBP and participant characteristics, injury history, training related factors, and work and personal factors were obtained.’, here, omit the words ‘were obtained’.

Thank you, this has been omitted. 

Introduction

To provide the most up-to-date picture of low back pain (LBP) prevalence in different sports, it would be beneficial to update the introduction and discussion sections on prevalence with findings from a recent study by Ansari and Sharma (2023) [Ansari, S., & Sharma, S. (2023). Prevalence and risk factors of chronic low back pain in university athletes: a cross-sectional study. The Physician and Sportsmedicine, 51(4), 361–370. https://doi.org/10.1080/00913847.2022.2108351] on chronic LBP of non-specific originin university athletes. Additionally, incorporating risk factors explored in this research would strengthen the introduction's overview of relevant studies on the topic.

Thank you for the suggestion. We have opted not to include this research study in the section on prevalence as we have placed the emphasis on high quality articles that have synthesised results that represent a wide range of athletic groups, ages etc. We don’t feel going specific to university-level athletes across 6 sports with the prevalence representing chronic LBP is appropriate here. Indeed, the prevalence reported in this paper is 7.7 to 15.6% which is considerably lower than the 44% in this study and 33-67% elsewhere. When discussing field-hockey specifically, we have referenced the only hockey papers available. 

In terms of the risk factors, it’s rather tricky to incorporate this into our study given the focus on chronic pain as noted above. It’s not clear if an association with chronic LBP can be extrapolated to make inferences for a single (or multiple) incidents of LBP over a 12-month period. For example, in Jonsdottir et al. (2019) study, they found that deprivation, gender and vigorous exercise was associated (positively or negatively) with chronic LBP but not acute LBP. Whilst other risk factors were more consistent for acute and chronic, the magnitude was substantially difference in some cases (e.g., 65+ years). As such, drawing comparisons between the study by Ansari and Sharma (2023) with our study could be problematic and best avoided in our opinion. 

Jonsdottir, S., Ahmed, H., Tómasson, K. and Carter, B. (2019). Factors associated with chronic and actue back pain in Wales: a cross-sectional study. BMC Musculoskeletal Disorders: https://bmcmusculoskeletdisord.biomedcentral.com/articles/10.1186/s12891-019-2477-4

There seems to be an error in citation number eight. The author's name is listed as Frett, but it should be Fett. Please update all references and the bibliography entry accordingly

Our apologies, this has now been corrected. 

Material and methods

It is recommended to use sub-headings in this section, example, ethical considerations and study registration, study setting and design, participants/sample size/eligibility criterion, data collection/questionnaire, statistical analyses, etc.

Thanks for the suggestion. We have now included three sub-heading to divide the Materials and Methods section. 

Were the data from the pilot testing version included in the present study?

Thanks for highlighting this. We can confirm that the data from the pilot was not included; this has now been made clear in the paper. 

What measures were used to depict the data, mean (SD)/ n (%), add details of these in the statistical analyses part.

This has now been included in the manuscript. We’ve added some additional information in the statistical analysis section and added “(n)” in Table 1

How were the variables fit into a univariate regression, was a significant correlation test was used to ascertain its inclusion in the univariate regression?

As we have taken a three-phrase approach to the regression, we did not determine if “significant” associations existed using a Chi Square test and Cramér’s V to construct the model. There were several reasons for this decisions:

1). We sought to take a theory-derived approach to the justification rather than purely statistical. As such, we allowed the associations to emerge without preliminary testing. 

2). The approach of running a univariable and multivariable model before constructing a final model is regarded as best practice and will likely control for redundant variables caused by high collinearity. We also considering this to be particularly suitable when considering the number of independent variables in this study. 

3). We did not wish to include or exclude variables based on associations and “significance” as this is recognised as limiting in exploring the association between risk factors and disease/pain/conditions etc. It is highly likely that this approach means removing variables at a univariable level despite potentially providing clinically useful information when placed in a multivariable or the final model. 

4). Some grouping such as playing position had small sample sizes and were slightly underrepresented. As such, this low statistical power in a simple test of association could result in misleading conclusions. 

Add details in the statistical analyses section on how the categorization of the variables was done for analyses.

We have included some insight on how the categories were formed in the ‘Data Collection and Questionnaire Development’ section and referred the reader to Table 1 rather than listing all of these in the Methods. We hope this is OK. 

Results

It is recommended to provide some of the basic demographic data in terms of mean (SD).

We cannot provide means and standard deviations as the data was collect as categorical data. As such, the descriptive data is included in-text or in table one as absolute numbers. 

Discussion

Lines 260-271: Add findings from the studies:

Moradi V, Memari A-H, and ShayestehFar M, et al. Low back pain in athletes is associated with general and sport specific risk factors: a comprehensive review of longitudinal studies. Rehabil Res Pract. 2015;2015 2015:850184 . DOI:10.1155/2015/850184.

Wilson F, Ardern CL, Hartvigsen J, et al. Prevalence and risk factors for back pain in sports: a systematic review with meta-Analysis. Br J Sports Med. 2021;55(11):601–607.

Thanks for the suggested papers. The top one has not been included as this doesn’t support provided any additional or robust evidence to support or refute the point being made in the paper. Also, we are not prepared to include a paper from a publishing company who has recently retracted 8000 papers due to their paper-mill problems. The paper by Wilson has been re-cited in this paragraph. 

Lines 318-323, To understand how sleep might influence LBP risk across different sports, it's worth including sport-specific results from the below mentioned study on sleep and LBP:

Ansari, S., & Sharma, S. (2024). Sleep Status and Chronotype in University Athletes with and without Chronic Low Back Pain: A Cross-Sectional Study. Sleep Science.

Thanks for the suggestion. We have not included this for the following three reasons: 

1). This study focused on chronic LBP which is not what has been captured in our study. 

2). Sleep quantity was not associated with chronic LBP as a single measure in this study. 

3). We only briefly referred to this as the OR crossed 1.0, so it was not a focus of our discussion given the mixed findings. The sentence noted focuses on occupational fatigue rather than sleep quantity. 

Reviewer #2: 

Abstract: in method, please state study design, sampling method. Include how many club involved for 194 participants

We have not provided the study design and sampling method. We are reluctant to include the number of clubs involved as we don’t know this accurately enough. Whilst we do know that 18 clubs agreed to share the questionnaire and participate, we did not collect club information to protect their anonymity. Therefore, it’s unclear to us if all 18 clubs are represented in the final 194 responses, and it wouldn’t be right for us to assume this. 

keyword: add low back pain

Low back pain is captured in the title, so we are unsure why this needs including as it doesn’t alter the discoverability of the work. 

Introduction:

line 100 after field hockey (ref)

Reference 6 and 7 have now been included. 

Materials and method:

Please add exclusion criteria

This has now been included. 

Please specify the statistical test

We might require some additional direction here as we have specified that a uni- and multi-variable logistic regression model within the statistical analysis section. If this point was referring to prevalence, we have now included the equation within the manuscript. 

Result: please include P value in Table 1

We have now included these as superscript numbers. 

Table 1= referent or reference?

We have opted to replace referent in the table with “ref” and then explain “ref = reference/referent category” to minimise any confusion. Our understanding is that both are acceptable within a logistic regression, but you’re correct to suggest reference is more commonly adopted. We hope this approach is acceptable. 

line 168-169 the sentence is not clear

We have rephrased this slightly to enhance the clarity. It now reads “Finally, 20% of our sample were, at the time of completing the questionnaire, competing as an international field hockey player.”

---

## [Editor Report · Decision Letter 1]

6 Jun 2024

Factors associated with non-specific low back pain in field hockey: a cross-sectional study of Premier and Division One players.

PONE-D-24-12011R1

Dear Dr. Dobbin,

We’re pleased to inform you that your manuscript has been judged scientifically suitable for publication and will be formally accepted for publication once it meets all outstanding technical requirements.

Kind regards,

Shazlin Shaharudin

Academic Editor

PLOS ONE
---

## [Editor Report · Acceptance letter]

18 Jun 2024

PONE-D-24-12011R1 

PLOS ONE

Dear Dr. Dobbin, 

I'm pleased to inform you that your manuscript has been deemed suitable for publication in PLOS ONE. Congratulations! Your manuscript is now being handed over to our production team.

Kind regards, 

on behalf of

Dr. Shazlin Shaharudin 

Academic Editor

PLOS ONE